# Evaluation of residue management practices on barley residue decomposition

**Grant Loomis**[1☯‡]**, Biswanath Dari** [2☯‡]*****, **Christopher W. Rogers**[3]**, Debjani Sihi**[4]

**1** University of Idaho Extension, Blaine County, Hailey, Idaho, United States of America, **2** Department of Crop and Soil Science, Oregon State University, Klamath Basin Research and Extension Center, Klamath Falls, Oregon, United States of America, **3** United States Department of Agriculture – Agricultural Research Service, Northwest Irrigation and Soils Laboratory, Kimberly, Idaho, United States of America, **4** Climate Change Science Institute and Environmental Sciences Division, Oak Ridge National Laboratory, Oak Ridge, Tennessee, United States of America

☯ These authors contributed equally to this work.
‡ These authors share first authorship on this work.
* b.dari@oregonstate.edu

**Data Availability Statement:** All relevant data are within the paper and its Supporting Information files.

**Funding:** CWR and BD No grant number - funded by study name, "Assessing Residue Source and

## Abstract

Optimizing barley (*hordeum vulgare* L.) production in Idaho and other parts of the Pacific Northwest (PNW) should focus on farm resource management. The effect of post-harvest residue management on barley residue decomposition has not been adequately studied. Thus, the objective of this study was to determine the effect of residue placement (surface vs. incorporated), residue size (chopped vs. ground-sieved) and soil type (sand and sandy loam) on barley residue decomposition. A 50-day(d) laboratory incubation experiment was conducted at a temperature of 25°C at the Aberdeen Research and Extension Center, Aberdeen, Idaho, USA. Following the study, a Markov-Chain Monte Carlo (MCMC) modeling approach was applied to investigate the first-order decay kinetics of barley residue. An accelerated initial flush of residue carbon(C)-mineralization was measured for the sieved (Day 1) compared to chopped (Day 3 to 5) residues for both surface incorporated applications. The highest evolution of carbon dioxide ($CO_2$)-C of 8.3 g kg$^{-1}$ dry residue was observed on Day 1 from the incorporated-sieved application for both soils. The highest and lowest amount of cumulative $CO_2$-C released and percentage residue decomposed over 50-d was observed for surface-chopped (107 g kg$^{-1}$ dry residue and 27%, respectively) and incorporated-sieved (69 g kg$^{-1}$ dry residue and 18%, respectively) residues, respectively. There were no significant differences in C-mineralization from barley residue based on soil type or its interactions with residue placement and size ($p$ >0.05). The largest decay constant k of 0.0083 d$^{-1}$ was calculated for surface-chopped residue where the predicted half-life was 80 d, which did not differ from surface sieved or incorporated chopped. In contrast, incorporated-sieved treatments only resulted in a k of 0.0054 d$^{-1}$ and would need an additional 48 d to decompose 50% of the residue. Future residue decomposition studies under field conditions are warranted to verify the residue C-mineralization and its impact on residue management.

Management Practices for Improving Fertilizer Recommendations in Cereal-based Cropping Systems" Idaho Barley Commission Website URL of funder: https://barley.idaho.gov/ NO.

**Competing interests:** The authors have declared that no competing interests exist.

## Introduction

Barley (*hordeum vulgare* L.) is an important cereal crop grown worldwide with a production of 142 million metric ton (MMT) in 2018–2019 crop year [1] with a total production of 3.09 MMT covering nearly 790,000 hectares in the USA [2]. Idaho represents one-third of total US barley production which accounted for ~1.1 MMT of barley production annually over 200,000 hectares [3]. Globally, fifty-nine countries import US barley largely for use in the beer industry [4]. Common practices of barley production involve harvesting of barley grains and afterwards, barley residue either remains in the field or is bailed and removed. Remaining residue in the field is an important source of soil organic carbon (SOC) [5]. In semi-arid regions of the Pacific Northwest (PNW) and western United States (for example, Idaho), soils are susceptible to soil organic matter (SOM) loss, thus, SOC levels may be low [6]. In addition to building SOC, residues reduce soil erosion, retain and recycle plant nutrients in the soil, and provide required energy for soil microbial processes [5,7–10].

Carbon to nitrogen (C:N) ratio is a contributing factor in determining the residue decomposition as affected by existing soil microbial communities. Sustainable nutrient management for barley production in Idaho and other parts of PNW are based on the knowledge of how residues affect the cycling of C and N, and the availability of crop nutrients in the soil. Critical factors in barley residue breakdown include soil type, tillage management, and management of the post-harvest barley residue. Barley residue can vary in its C: N ratio (54 to 80:1) depending on the growing condition and location [11]. Further, tillage operations in Idaho and other PNW regions vary widely from no-tillage, direct-seed operations to intensive conventional tillage operations that can subsequently affect residue breakdown. Conventional tillage incorporates residues, surface soil is turned over into the sub-surface soil [12] and the rhizosphere becomes more stable for residue breakdown [13,14]. Incorporation of crop residues in a wheat (*triticum aestivum L.*) field in Oregon (~PNW) over a 26-month field study showed that the net mineralization was typically greater when residues were plowed into the soil (total 85%) than when residue was left on the soil surface (25% and 31%, respectively) [15] over the course of experiment, and provided greater nutrients for subsequent crops. However, the opposite trend of increased breakdown of corn (*zea mays L.*) residue occurred when residues remained on the surface of the soils (~no tillage) compared to incorporated in recent studies in Iowa [10,16]. The method of residue placement (surface vs. incorporated) and consequently, the contact between the soil and the residue have had conflicting results indicating the effects of other confounding factors (i.e. residue properties, native microbial communities) on the residue breakdown rate in previous research. Residue particle size of barley residue in the field as part of management practices could aid either faster or slower breakdown by various mechanisms [14]. In general, finer sized residue would allow more surface area for microbial activities and nutrient and/or water retention in the surface of the soil to facilitate residue breakdown [5,10]. In contrast, the application of residues in coarser chopped conditions would result in reduced surface contact between the soil and barley straw which has been claim to slow down the rate of residue breakdown by microbial decomposers [17]. Thus, smaller sized residues may favor C-mineralization and residue breakdown in barley by either increased microbial activities particularly by increasing soil temperature [15] or by filling the soil pore-space in higher magnitude which facilitate decomposition [5]. Alternately, an increased C-mineralization (~8% more) from small-grain residues with the application of coarser-sized residue as compared to fine-sized residue during a 65-d laboratory incubation study at 25°C were also reported [11].

Residue breakdown is also affected by soil type and associated soil properties (for example, organo-mineral interactions) specific to a region as well as interactions that occur in the

rhizosphere among the plant, soil microbial organisms, and soil flora and fauna [18]. Barley residue decomposition was affected by soil texture, for example reduced breakdown with a greater adsorption of organic N and C (stronger organo-mineral bonding) in clay soils [18] vs. greater breakdown with a reduced rate of SOM decomposition in highly aerated sandy loam soils [13,19]. Soil type has been reported to have a more prominent effect on the rate of decomposition than the residue N content [13]. Recently, no differences in decomposition rate of corn residues over an extended period of time were observed in a silty-clay loam soil in Iowa after addition of N to the soil [5]. Other associated soil properties such as soil moisture in addition to soil type significantly influences the breakdown of the residue [13].

The effects of various factors in the process of barley residue decomposition as mentioned above showed mixed results based on the literature available. Additionally, the effect of post-harvest management practices of barley residues in irrigated production systems on C-mineralization needs to be quantified and understood to ensure sustainable barley production. Achieving sustainable barley production through understanding the dynamics of residue carbon breakdown and residue $CO_2$-C loss for subsequent crop production is the focus of this study. Our objectives were (i) to determine the effect of residue placement (surface vs. incorporated), residue size (chopped vs. sieved-ground) and soil type (sand and sandy loam) on barley residue decomposition rate, (ii) to quantify the barley residue decomposition with a modeling approach to guide proper resource management for barley and cereal production.

## Materials and methods

### Site description

Two soil samples (with four field replicates) were collected from a common soil series (Declo Loam; Coarse-loamy, mixed, superactive, mesic Xeric Haplocalcids) in southeastern Idaho [20,21] (Table 1). While both the soil samples were from the same soil series they differed in textural class (sand versus sandy loam) [21]. Before soil sample collection, the lands were cultivated with barley (Hordeum vulgare, *L*.). The average soil organic carbon content measured by loss on ignition (LOI) in the study region was 2 g kg$^{-1}$ [22]. Within a sampling location, an approximately 0.1 ha area was sampled by collecting and compositing 20 sub-samples using a 7.6-cm bucket auger from depths of 0- to 15- cm. Tillage operations can be highly varied in Idaho and other western States. The depth of 0–15 cm is an approximation for shallow tillage operations that are conducted using disc plows [5]. Collected soil samples were dried in a forced-convection oven at 40˚C, and subsequently crushed and passed through a 2-mm sieve.

**Table 1. Basic properties of soil used in the laboratory incubation study conducted at the Aberdeen Research and Extension Center, Aberdeen, ID, USA.**

| Properties | Sand | Sandy Loam |
|---|---|---|
| **Sand (g kg$^{-1}$)** | 898 (±4.9)[†] | 661 (±5.6) |
| **Silt (g kg$^{-1}$)** | 57 (±15.1) | 239 (±5.6) |
| **Clay (g kg$^{-1}$)** | 45 (±11.0) | 100 (±0.02) |
| **pH** | 8.1 (±0.01) | 8.4 (±0.02) |
| **EC (μS cm$^{-1}$)** | 102 (±3.04) | 148 (±5.9) |
| **SOM (g kg$^{-1}$)** | 9 (±0.6) | 15 (±0.6) |
| **Total N (g kg$^{-1}$)** | 0.46 (±0.01) | 0.69 (±0.02) |

[†]values in the parenthesis indicate standard errors.

EC; electrical conductivity, SOM; soil organic matter; N; Nitrogen.

## Soil description

Soil particle size analysis was performed using the hydrometer method (Table 1) [23]. Soil pH and electrical conductivity (EC) were determined potentiometrically using a 1:1 soil to deionized water ratio [23] using a soil pH meter (Orion Star[AM] A215 pH/Conductivity Benchtop Multiparameter meter, Thermo Fisher Scientific Inc., Waltham, MA, USA). The loss on ignition (LOI) analysis was conducted on samples where 10 g of the sample was dried at 105 ˚C for 2 h and placed in a desiccator for 1 h. Samples were then combusted in a muffle furnace at 360 ˚C for 2 h, dried for 1 h at 105 ˚C and equilibrated in a desiccator for 1 h. The SOM content was determined by LOI based on the difference in initial and final weights [23,24]. Total nitrogen (TN) was measured by high-temperature combustion using a VarioMax CN analyzer (Elementar Americas, Inc. Mt Laurel, NJ).

## Residue description

The residue used in the experiment was obtained from a common malt barley cultivar (Harrington) which is grown for malt production in the study region [25]. Residues were collected in 2017 after harvest of a barley crop grown under standard production practices at the Aberdeen Research and Extension Center, Aberdeen, ID [26]. An estimated dry matter production for barley excluding grain (~8000 kg ha$^{-1}$) was chosen as the rate of residue added for the study [27]. The residue was characterized for cellulose, hemicellulose, lignin and ash content. The cellulose–Acid Detergent Fiber (ADF; ash-free) content were determined using the method of 'Fiber (Acid Detergent) and Lignin in Animal Feed: Section: 973.18' [28] with a modification of using Whatman 934-AH glass micro-fiber filters with a capacity of 1.5μm particle retention instead of using the fritted glass crucible. The hemicellulose-Neutral Detergent Fiber (NDF; ash-free) content was determined using the method described by [29]. Lignin, a critical factor influencing digestibility of the plant cell wall, was analyzed by the method outlined by [30]. Followed by fiber (ADF and NDF) and lignin analyses, the ash content in the residue samples were analyzed using the methods detailed in the 'Ash of Animal Feed; section: 942.05' [28].

## Laboratory incubation experimental approach

A 50-d laboratory incubation experiment was conducted at a constant temperature (25˚C) and moisture (60% water-filled pore space). Although we did not mimic the field condition, the experimental set-up we used allowed us to compare our findings with other microcosm studies focused on residue decomposition [5,31]. The experiment was arranged as a randomized complete block (RCB) design with two types of residue placement (surface vs. incorporated), two residue sizes (chopped vs. sieved), and two soil textures (sand vs. sandy loam) with four replicates for each. Barley residue (4.1 g) and soil (100 g) were placed in a Mason jar. A 500 mL capacity wide-mouth mason jars with an 86 mm dome lid modified to support a 60-mm petri dish [32] were used in this study. The inner diameter width 76 cm, outer diameter width of 82 cm, and height of 113cm. (The inner diameter width 76 cm, outer diameter width of 82 cm, and height of 113cm). Surface applied residue was evenly spread on the surface of the soil and incorporated residue was thoroughly mixed into the soils. Residue size included chopped (~7-cm) and sieved (ground to pass a 2-mm sieve) treatments.

The study was conducted based on incubation methods described by [5]. A 60 mm Petri-dish containing 5 mL of 1 M NaOH solution was placed in each Mason jar to capture $CO_2$ evolved from the soil-residue mixture during the incubation. The amount of $CO_2$ evolved from residue decomposition trapped in NaOH solution at specific time intervals was determined by titration immediately by adding 5 mL of 2 M barium chloride ($BaCl_2$) solution and 2 to 3 drops of phenolphthalein indicator to each petri-dish and pH endpoint titrating with 1 M

HCl solution using a digital auto-titrator (848 Titrino Plus, Metrohm Lts., Herisau, Switzerland) until the pH endpoint was reached. A new petri dish with 1 M NaOH was used at each time interval after taking the previous petri-dish from individual mason jars. Three mason jars containing only soil of each type without any residue added were used as a control to monitor $CO_2$ release from the soil. Additionally, three empty Mason jars without any soil or residue were included as an experimental control (control Mason jars) which were used to calculate the total $CO_2$-C evolved from each treatment in the headspace (See the calculation section in supporting information).

Mason jars were weighed initially with the soil-residue mixtures and each Mason jar was weighed each time the petri-dish was changed for taking the reading. Additionally, the amount of $CO_2$ evolved from each jar via absorption by sodium hydroxide (NaOH) in the petri-dish was taken into account for weight reduction in each jar. To maintain the constant soil moisture, the overall reduction in the weight of the Mason jar was compensated by sprinkling the exact amount of deionized water in each jar during the entire experiment.

## Calculations

First-order decay constant and decay time. Once the endpoint in titration was achieved, the amount of $CO_2$ retained in each petri-dish was determined by using the formula described by [33]. The initial rate of residue decomposition or decay constant (k) was calculated using a simple first-order decay function (Eq 1, see [34] for more details) within a Bayesian Markov Chain Monte Carlo (MCMC) framework (Eq 1).

$$C_t = C_0(1 - e^{-kt}) \tag{1}$$

Where,

$C_t$ = C content at time t (day)
$C_0$ = initial C content (mg)
k = first-order rate constant ($day^{-1}$)
t = time (day)
The joint probability distribution of the Bayesian MCMC model used is as follows (Eq 2):

$$[C_0, k, \sigma_p^2 | C_t] \propto \Pi_{i=1}^n \text{gamma}\left(C_{ti} \Big| \frac{(C_0 * (1 - e^{-kt_i}))^2}{\sigma_p^2}, \frac{(C_0 * (1 - e^{-kt_i}))}{\sigma_p^2}\right) \times \text{beta}(k|\alpha, \beta) \times$$
$$\text{gamma}(C_0|\alpha, \beta) \times \text{gamma}(\sigma_p|\alpha, \beta) \tag{2}$$

Where, $C_0$, $C_t$, and $\sigma_p$ (aka process model uncertainty) followed uninformed gamma prior distributions and k followed an uninformed beta prior distribution.

We implemented the MCMC method using the *rjags* package in R (version 3.4.3). Posterior distributions of k were estimated with 10,000 model iterations after discarding the first 5000 runs. Model convergence was diagnosed with the Gelman–Rubin diagnostics [35] and model suitability were determined with a Bayesian p value.

Using decay constant (k) values, time required for 50, 75, and 99% residue C-mineralization from barley residue was estimated (Eqs 3, 4 & 5) as follows:

$$t_{0.50} = \ln(2/k) \tag{3}$$

$$t_{0.75} = \ln(4/k) \tag{4}$$

$$t_{0.99} = \ln(100/k) \tag{5}$$

## Calculations of $CO_2$-C decomposed

Once the endpoint in titration was achieved, the amount of $CO_2$ retained in each petri-dish was determined by using the following formula (33; Eq 6).

$$CO_2 = (X - Y) \times N \times W \tag{6}$$

Where,

X = volume of acid needed to titrate the petri-dish solution from the Mason jars with soil sample only to the end point i.e. 'blank', (mL)

Y = volume of the acid needed to titrate the petri-dish solution from the Mason jars with soil-residue sample to the end point i.e. 'blank', (mL)

N = normality of the acid, $(mL^{-1})$

W = the equivalent weight of C in $CO_2$; W would be 6 if data is to be expressed in terms of C, (mg $CO_2$-C).

## Statistical analyses

Treatment effects on cumulative C-mineralization rate (residue breakdown), the percentage of residue decomposed, first-order decay kinetic parameters such as decay constant and the number of days to decompose the residue was analyzed using an analysis of variance (ANOVA). Statistical analyses were performed in JMP 13.0 (SAS Institute, Cary, NC, USA). Tukey's multiple comparison procedures, as well as the corresponding letter grouping method, was used to separate the treatment means as post-hoc multiple comparisons where $p \leq 0.05$.

# Results

## Characterization of residue

The barley residue had a TN of 4.6 g $kg^{-1}$ dry residue whereas, the TC was 387.5 g $kg^{-1}$ dry residue which gave a C:N ratio of 85:1 (Table 2). In general, a higher C:N ratio is expected for barley residue as a cereal crop compared to broadleaf crops which matched our data. The cellulose and hemicellulose content of the barley residue were 447 and 670 g $kg^{-1}$ dry residue, respectively, where the lignin content was 64 g $kg^{-1}$ dry residue.

## Daily and cumulative decomposition of barley residues

The daily response of the C-mineralization rate (as measured by $CO_2$ released) with respect to the residue placement (surface vs. incorporated) and residue size (chopped vs. sieved) were variable. Both the soils showed a similar pattern in daily residue decomposition breakdown (Fig 1A and 1B). In general, an initial C-mineralization flush by individual treatment combinations for both soils ranged between 2 to 5 days (Fig 1A and 1B). However, the accelerated initial mineralization rate at the beginning of the experiment was obtained quicker and at a

**Table 2. Basic properties of barley[†] residue samples used in laboratory incubation study conducted at the Aberdeen Research and Extension Center, Aberdeen, ID, USA.**

| Properties | Unit (on dry residue weight basis) | Average |
|---|---|---|
| Total N | g kg[-1] | 4.6(±0.9) [††] |
| Total C | g kg[-1] | 387.5(±4.9) |
| C:N | - | 85(±1.7) |
| Cellulose | g kg[-1] | 447(±16) |
| Hemicellulose | g kg[-1] | 670(±16) |
| Lignin | g kg[-1] | 64(±2.0) |
| Ash | g kg[-1] | 120(±8.0) |

[†]barley cultivar: Harrington

[††]values in the parenthesis indicate standard errors.

N; nitrogen, C; carbon, C: N; carbon to nitrogen ratio.

greater magnitude for the sieved residues (at Day 1 to 2) than chopped (at Day 3 to 5) for both surface application and incorporation. The highest evolution of $CO_2$-C of 8.3 g kg[-1] dry residue was observed on Day 1 from the incorporated-sieved application for both the soils (Fig 1A and 1B). The cumulative decomposition of barley residue increased until it reached a plateau for both the soils (Fig 2A and 2B. The pattern of this cumulative decomposition over a period of

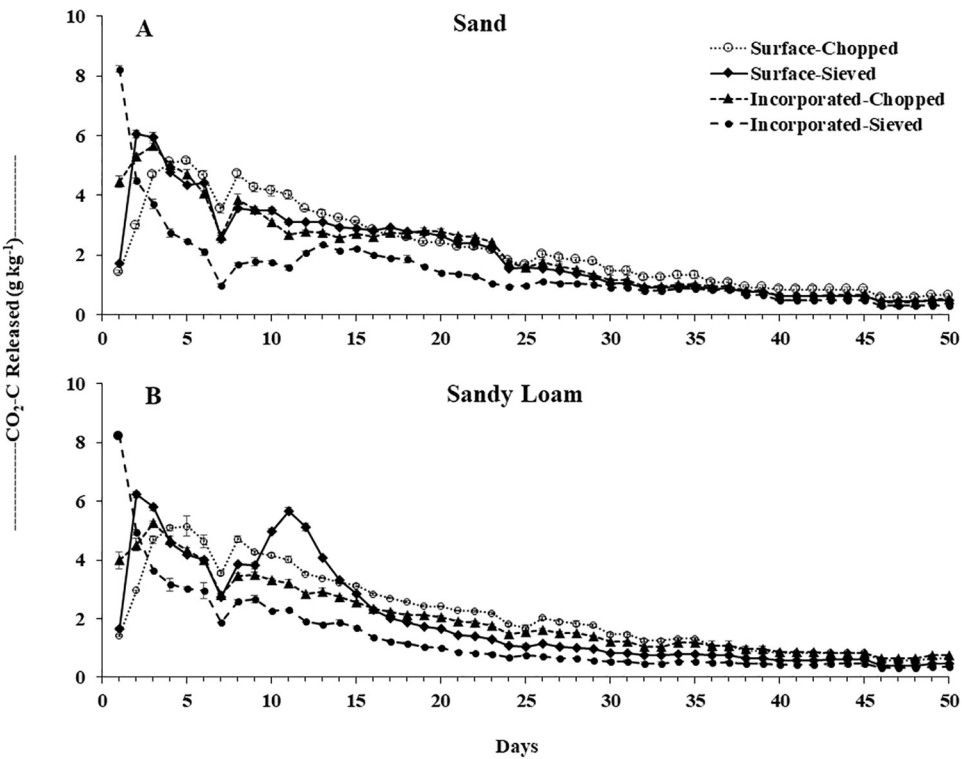

**Fig 1. Amount of $CO_2$-C released (rate as calculated per kg of residues) on daily basis for barley residue decomposition under residue placement and residue size treatment combinations for A) sandy and B) sandy loam soils in 50-days' laboratory incubation study at 25˚C.**

50-d followed a similar trend for all the treatment combination as daily C-mineralization rate for both soils (Fig 2A and 2B).

## Effects of residue management on barley residue decomposition

The ANOVA results indicated that residue placement, residue size, and their interaction had a significant effect on cumulative $CO_2$-C release, percent of residue decomposed, and calculated residue decay constant over a period of 50-d ($p<0.001$) (Table 3). There were no significant effects of soil type (sandy vs. sandy loam) on any of the measured output in our study although two soils were textually and chemically different. Similarly, the higher-order interaction among residue placement, residue size, and soil types were all non-significant for cumulative $CO_2$-C release, decay constant, and percent of residue decomposed. Averaged across soil types, the highest (107 g $kg^{-1}$ dry residue) and lowest (69 g $kg^{-1}$ dry residue) amount of cumulative $CO_2$-C released over 50-d were from surface-chopped and incorporated-sieved treatments, respectively (Fig 3). However, there were no significant differences in $CO_2$-C release between surface-sieved (101 g $kg^{-1}$ dry residue) and incorporated-chopped (98 g $kg^{-1}$ dry residue) treatments (Fig 3A).

As expected, the percentage of residue decomposed followed the same pattern as the C-mineralization rate for various treatment combinations (Fig 3B). When averaged across soil types, the highest and lowest amount of residue decomposed over a 50-d decomposition period was measured in surface-chopped (27%) and incorporated-sieved (18%), respectively (Fig 3B). The surface-sieved vs. incorporated-chopped treatments showed no significant

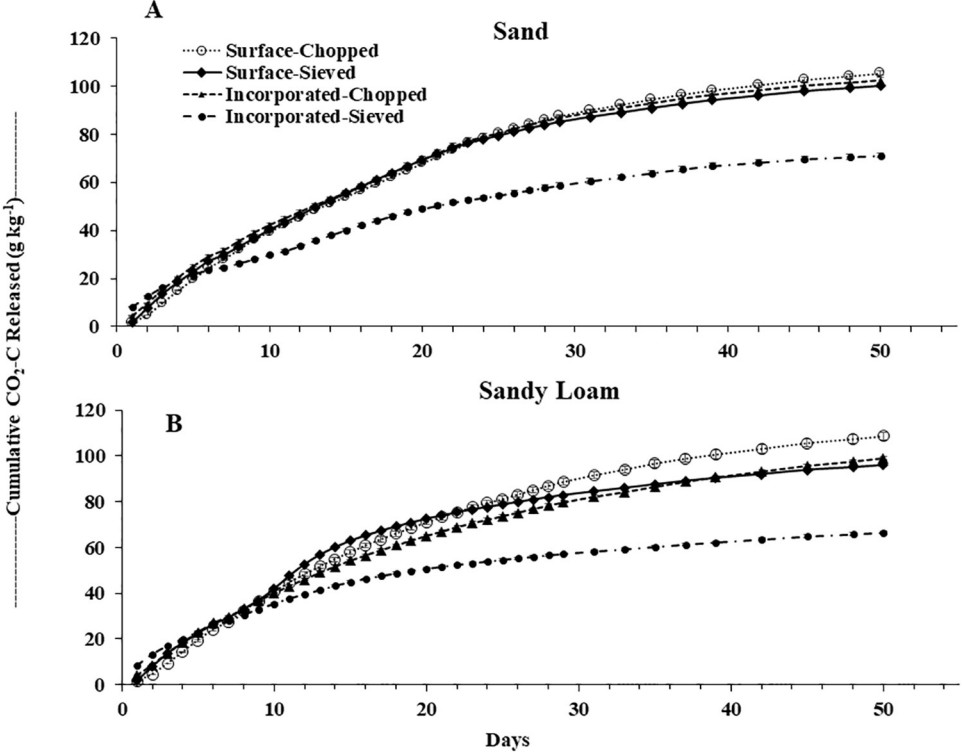

**Fig 2. Amount of cumulative $CO_2$-C released (rate as calculated per kg of residues) for barley residue decomposition under residue placement and residue size treatment combinations for A) sandy and B) sandy loam soils in 50-days' laboratory incubation study at 25˚C.**

**Table 3. Analysis of variance (ANOVA) P values for cumulative decomposition rate, percentage residue decomposed, decay constant (k) and associated parameters over 50-days' laboratory incubation study for barley residue decomposition conducted at the Aberdeen Research and Extension Center, Aberdeen, ID, USA.**

| Sources of variation | Cumulative $CO_2$-C decomposition | Residue decomposed | Decay constant (k) | $t_{(0.5)}$ | $t_{(0.75)}$ | $t_{(0.99)}$ |
|---|---|---|---|---|---|---|
| **Residue placement (RP)** | <0.001 | <0.001 | <0.001 | <0.001 | <0.001 | <0.001 |
| **Residue size (RS)** | <0.001 | <0.001 | <0.001 | <0.001 | <0.001 | <0.001 |
| **Soil type (ST)** | 0.09 | 0.09 | 0.34 | 0.07 | 0.23 | 0.09 |
| **RP x RS** | <0.001 | <0.001 | <0.001 | <0.001 | <0.001 | <0.001 |
| **RP x ST** | 0.17 | 0.16 | 0.07 | 0.18 | 0.09 | 0.20 |
| **RS x ST** | 0.10 | 0.10 | 0.66 | 0.13 | 0.72 | 0.12 |
| **RP x RS x ST** | 0.24 | 0.27 | 0.09 | 0.35 | 0.10 | 0.41 |

differences in their mean values of percentage of residue decomposed (25% for both the treatments).

## First order decay kinetics and half-life of barley residues

The ANOVA results indicated that the residue placement, residue size, and their interaction have a significant effect on the decay constants, $t_{(0.5)}$, $t_{(0.75)}$ and $t_{(0.99)}$ residue C-mineralized estimated ($p<0.001$) (Table 3). There were no significant effects of soil types on any of the calculated parameters. The first-order decay kinetics parameters calculated using MCMC modeling approach were well aligned with model parameter uncertainty. In addition the sensitivity analyses were performed and model predictions were satisfactory as shown in (S1 Fig). The highest k value ($0.0083\pm0.0005$ $d^{-1}$; $R^2 = 0.997$) with a predicted residue half-life to mineralize barley residue ($t_{(0.5)}$) of about $80+3$ d was obtained for surface-chopped applied residues when averaged across soils (Table 4). This value did not statistically differ from surface-sieved (k = $0.0080\pm0.0006$ $d^{-1}$; $R^2 = 0.993$) and incorporated-chopped (k = $0.0079\pm0.0007$ $d^{-1}$; $R^2 = 0.993$) treatment combinations. The lowest k value ($0.0054\pm0.0005$ $d^{-1}$; $R^2 = 0.988$) was calculated for incorporated-sieved residues when averaged across soils types (Table 4). Incorporated-sieved treatments would require the maximum duration to decompose 50% of the residue (~128 d). The k and $t_{(0.5)}$ values of incorporated-sieved residues differ significantly from other three treatment combinations (i.e. surface-chopped, surface-sieved, and incorporated-chopped treatments). Similar trends were found for $t_{(0.75)}$ and $t_{(0.99)}$ for all the treatment combinations (Table 4).

## Discussions

### Daily and cumulative decomposition of barley residues

The initial flush from C-mineralization for crop residues under laboratory, greenhouse, or field conditions is usually significantly different from the decomposition rate for the rest of the experimental duration [5,10,11,36]. This is because the maximum decomposition rate occurs in this period due to soil nutrient dynamics, moisture availability (and soil aeration status), and labile organic material that first undergo decomposition by triggering microbial activity. The initial higher flush of barley straw breakdown or greater C-mineralization in our study was also observed in other cereal residue decomposition studies [5,10,11,31,36].

The rapid rate of C-mineralization from residues in the initial phase (Day 1 to 2) was possibly due to a higher surface area and associated stimulation in microbial activities in sieved compared to the chopped residues (Fig 1A and 1B). Our results were in agreement with the findings by [36] who showed the greater mineralization by finer sized wheat straw at the initial

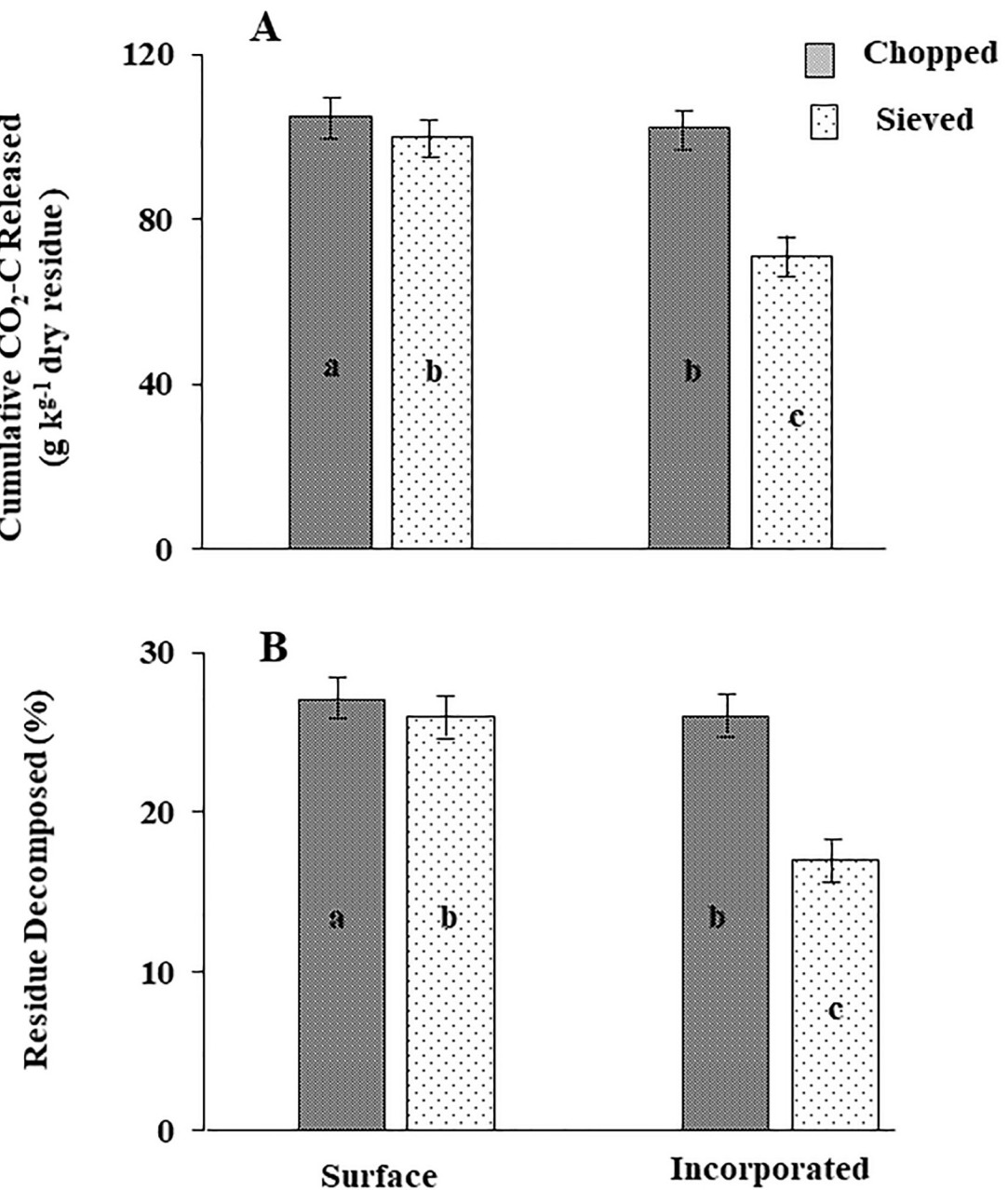

**Fig 3. Effects of residue placement and residue size on A) cumulative $CO_2$-C released (rate as calculated per kg of residues) and B) percentage of residue decomposed for barley residue decomposition averaged over two soils types in 50-days' laboratory incubation study at 25 ˚C.**

phase of residue addition (~1–2 days). After the initial transitory phase, the decreased rate of decomposition for surface-sieved residues might be attributed due to the exhaustion of those easily decomposable C sources and possibly due to the residue-mineral interaction later in the experiment [11]. They noted a similar pattern of decomposition where finer sized rye and wheat residues (0.05- to 1-mm sieved particles of residues) produced less $CO_2$-C than larger sized residues (7-mm) in their experiment when mixed with a silty soil at 25˚C in a laboratory incubation.

**Table 4. First-order decay model parameters for sequential barley residue decomposition under residue placement and residue size treatment combinations averaged across soil types in 50-days' laboratory incubation study at 25˚C.**

| Residue placement | Residue size | Mean k-values (d⁻¹) (±95% CI) | Goodness of fit statistics ($R^2$) | Mean $t_{(0.5)}$ (±95% CI) | Mean $t_{(0.75)}$ (±95% CI) | Mean $t_{(0.99)}$ (±95% CI) |
|---|---|---|---|---|---|---|
| **Surface** | **Chopped** | 0.0083a[††] (±0.0005) | 0.997 | 80b (±3) | 165b (±6) | 552b (±40) |
| | **Sieved** | 0.0080a (±0.0006) | 0.993 | 86b (±5) | 171b (±10) | 569b (±34) |
| **Incorporated** | **Chopped** | 0.0079a (±0.0007) | 0.993 | 88b (±8) | 175b (±16) | 582b (±53) |
| | **Sieved** | 0.0054b (±0.0005) | 0.988 | 128a (±13) | 257a (±22) | 853a (±72) |

Different letters for each parameter indicate significant differences between residue management, residue size and their interaction, as compared using Tukey's Protected honest significant difference (HSD) test at $p < 0.05$, respectively. CI; confidence interval; SL; sandy loam soils.

Enhancing the physical contact between crop residues and soil minerals may have further promoted the residue-mineral interactions, resulting in lower cumulative decomposition in the incorporated treatment than the surface treatment [5,11,37]. The rate of residue decomposition, subsequently, declined after approximately Day 5 regardless of treatment combination for both the soils (Fig 1A and 1B). This pattern followed other studies conducted either in field or laboratory with residues from different cereal crops grown in variable soil types [5,10,11,31]. These decreased rates of $CO_2$ evolution from the residues were followed by a slight increase at Day 8 to 9 for all treatments and again decreased until the end of the experiment. The difference in the decomposition rate among treatment combinations remained consistent except for the surface applied sieved residues which had a peak $CO_2$ rate of 5.8 g kg⁻¹ dry residue on Day 11.

## Effect of residue management on barley residue decomposition

Our study indicated that during the duration of the experiment the application of chopped residues on the surface of soils (Fig 3A and 3B) representing a no-tillage condition, promotes residue breakdown slightly faster than incorporated-chopped residue representing the conventional tillage practices and is supported by other findings where a similar pattern was observed [10,38]. Although no microbial community analyses were performed under the present study, we visually observed greater fungal growth on the surface applied residue than incorporated. Thus, different microbial interactions with the soil-residue mixture could be expected with incorporated residues as compared to surface residues. Our findings were also supported by [38], that reported a reduced rate of residue breakdown due to the lack of fungal communities when sorghum (*sorghum bicolor* L.) residues were incorporated as compared to the residue that was left on the surface of the soil. The results from our study could have value in promoting conservation practices and preserving residues on the soil surface rather than incorporation via conventional tillage practices in the PNW where soil erosion is a reasonable concern for soil and water quality maintenance. This is worth mentioning that sieved residue had higher decomposition rate than chopped residue in the initial stage, which may be due to higher surface area and thus more access for microbial reaction. However, the trend in the later stage of decomposition phase (after ~Day 5) reverse.

Our results provide evidence that the physical size of the barley residue (chopped vs. sieved) had a significant effect on the C-mineralization rate. For example, the surface-chopped and incorporated-chopped residue produced significantly greater $CO_2$-C than surface-sieved and incorporated-sieved residues, respectively (Fig 3A). The surface-chopped residues were more conducive to decomposition (decomposed 27% of applied residue) and produced greater amount of $CO_2$-C than surface-sieved residues over a 50-d period. The sieved residues may

have been stabilized by association with soil mineral surfaces, rendering them inaccessible to decomposition as compared to the sieved residues [11,37,39]. Differences were observed in the initial decomposition rate through day-5 which afterwards disappeared. In addition, the amount of C that were not mineralized in the initial period decomposition (~ 5 days) remained quite stabilized in the soil for longer duration after initial flush of $CO_2$-C release. Thus, the fine-sized grinding of residue provided no advantage over the chopping of residues in terms N release (and subsequent crop uptake) from decomposing residue for sustainable agricultural production [11]. We conclude that extreme mulching practices did not have an advantage in terms of C-mineralization and residue breakdown. Our results were supported by other field studies which showed greater cereal reside decomposition was observed in surface-chopped applied residue by maintaining a larger particle size of residues in the field [10,11,31,36].

Our study did not show any significant differences in the cumulative $CO_2$-C release by the two soils (Table 3). These two soils possess similar soil texture, however, these soils were selected as they are representative of the relatively similar soils found in the study region (south-eastern Idaho) (Table 1). Due to their similar soil texture, the $CO_2$-C mineralized from both the soils irrespective of the type of residue added were similar. This might be due to the fact that both soils have lower clay or organic matter content (0.9–1.5%) which has influenced the aeration and moisture status of the soil-residue mixtures [18]. Further, the presence of clay mineral (organo-clay mineral association) controls the magnitude of residue-mineral interactions which was not significantly different between these two soil types as both are relatively low in clay content. Additionally, other associated soil biological factors such as SOM did not vary widely which could have possibly made a difference in microbial populations and nutrient dynamics in these two soil types, and thus, differences in $CO_2$-C release from the soils.

## Prediction of barley residue decomposition: Modeling approach

The first-order kinetic model parameters were calculated using the MCMC approach described the C-mineralization during the initial stages of barley residue decomposition (Table 4). Our study indicated that the highest and lowest rate of both C-mineralization and percentage of residue decomposed were reported from surface-chopped and incorporated-sieved residues, respectively. This resulted in higher values of k for surface-chopped residues, which did not differ from surface sieved or incorporated chopped, as compared to the lowest k values in incorporated-sieved residues. It implies that the incorporated-sieved residues would take the longest time to decompose 50%, 75%, and 99% of the residues and only 18% of the residues were decomposed under the course of the experiment (50-d) (Table 4). Further microbial community analyses which estimates the population of heterotrophic microorganisms (e.g. fungi, different bacterial phylum, etc.) and information regarding their community composition and structure (e.g. bacterial:fungal ratio determination using Phospholipid-derived fatty acid; PLFAs [40]) across different soil types might explain the variability in k values for various treatment combinations.

The implication of the calculation of decay constant and predicted number of days to decompose a certain percentage of residue have serious consequences on residues remaining in the field. For example, if we aim towards a residue management practice that will facilitate residue decomposition and increase the nutrient availability especially mineral C and N to subsequent crops. However, faster residue mineralization might have a priming effects by increasing the decomposition of native organic matter, thus, reducing the soil's organic matter buildup [41] The enhancement in the native SOM by addition of fresh substrate in the initial stage of this study was driven by a temporary increase in the biomass feeding on the more

labile soil C [11]. However, to capture the priming effect, it would require labeling of either soil or the residue, which warrant a potential future study.

The most critical factor in soil C sequestration in this type of study is the enhanced physical contact between decomposing residues and soil minerals or organic-mineral contacts. For example, surface application of crop residues can reduce soil disturbance and increase SOC content but heavily respired by microbial communities, thus potentially contributing most to the formation of SOC [11]. On the other hand, incorporated residue in the soil can potentially enhance the SOC stock by having limited physical contact between residues and soil minerals [37]. Thus, the difference between the C stabilized from residue incorporation and C mineralized from tillage determines the net effect on the SOC stocks. These results can suggest that enhancement of physical contact/protection between organic materials and soil minerals (and/ or organo-mineral interaction) may promote C stabilization as determined by various treatment combinations in our study. To that end, we need to realize that two potential strategies need to be considered for soil C stock in agricultural soils such as no-tillage agricultural practices and crop residue or manure incorporation. Therefore, a better understanding of mechanisms that control organic-mineral interactions is needed for us to ameliorate current agricultural practices that facilitate SOC retention and sequestration.

## Conclusions

Our study considered some critical factors in barley post-harvest residue breakdown such as residue application methods, residue, size and soil types. This 50-d laboratory incubation study resulted in the C-mineralization rate of barley residues in the following order: surface-chopped>surface-sieved = incorporated-chopped>incorporated-sieved. Results indicated the magnitude of mineralizable C released from residue decomposition under different treatments were variable. Additionally, the effects of no-till and conventional tillage as represented by surface vs. incorporated residue application methods on barley residue decomposition under laboratory conditions were documented. Thus, the relevance and application of our findings would be critical in providing practical information on residue decomposition of a barley crop for various residue application methods. Although our study indicated that it is possible to obtain a greater residue C-mineralization for surface-chopped residue than surface sieved under controlled laboratory conditions, it may impact SOM build up. However, barley residue decomposition study under field conditions along with microcosm experimentation in laboratory, and measured at varying times during the crop growing season is warranted to achieve a definite conclusion. Besides, the inclusion of residues in soils with varying SOM content and/ or texture profile (e.g. fine- vs. coarse-textured) would be helpful to achieve a definite conclusion. As recommendations are developed, they should consider the advantages and disadvantages of faster or slower residue breakdown under field condition, which is location- and soil-specific. Thus, it would be interesting to see if various residue management practices have coupling effects on soil C (soil C formation and loss) and N cycle (nitrification and denitrification) processes.

## Supporting information

**S1 Data.**
(XLSX)

**S1 Fig. Calibration and prediction of the first order decay constant for residue decomposed under barley residue decomposition study conducted in laboratory at 25˚C.**
(DOCX)

## Acknowledgments

The help and support from Scott Pristupa and Irene Shackelford in the laboratory are greatly acknowledged. The author would also like to acknowledge Dr. Mahdi Al-Kaisi from Iowa State University for all his support and suggestions during the entire experiment.

## Author Contributions

**Conceptualization:** Grant Loomis, Biswanath Dari, Debjani Sihi.

**Data curation:** Grant Loomis, Biswanath Dari, Debjani Sihi.

**Formal analysis:** Grant Loomis, Biswanath Dari, Debjani Sihi.

**Funding acquisition:** Christopher W. Rogers.

**Investigation:** Christopher W. Rogers.

**Methodology:** Grant Loomis, Biswanath Dari, Debjani Sihi.

**Project administration:** Biswanath Dari, Christopher W. Rogers.

**Resources:** Grant Loomis, Christopher W. Rogers.

**Software:** Biswanath Dari, Christopher W. Rogers, Debjani Sihi.

**Supervision:** Biswanath Dari, Christopher W. Rogers.

**Validation:** Biswanath Dari, Christopher W. Rogers, Debjani Sihi.

**Visualization:** Grant Loomis, Biswanath Dari, Debjani Sihi.

**Writing – original draft:** Grant Loomis, Biswanath Dari, Debjani Sihi.

**Writing – review & editing:** Grant Loomis, Biswanath Dari, Christopher W. Rogers, Debjani Sihi.

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
