## [Decision Letter · Decision Letter 0]

4 Feb 2020

PONE-D-19-33598

Evaluation of residue management practices on barley residue decomposition

PLOS ONE

Dear Dr. Dari,

Thank you for submitting your manuscript to PLOS ONE. After careful consideration, we feel that it has merit but does not fully meet PLOS ONE’s publication criteria as it currently stands. Therefore, we invite you to submit a revised version of the manuscript that addresses the points raised during the review process.

We would appreciate receiving your revised manuscript by Mar 20 2020 11:59PM. To enhance the reproducibility of your results, we recommend that if applicable you deposit your laboratory protocols in protocols.io, where a protocol can be assigned its own identifier (DOI) such that it can be cited independently in the future. For instructions see: http://journals.plos.org/plosone/s/submission-guidelines#loc-laboratory-protocols

We look forward to receiving your revised manuscript.

Kind regards,

Vassilis G. Aschonitis

Academic Editor

PLOS ONE

Reviewers' comments:

Reviewer's Responses to Questions

**Comments to the Author**

1. Is the manuscript technically sound, and do the data support the conclusions?

Reviewer #1: Yes

Reviewer #2: Yes

2. Has the statistical analysis been performed appropriately and rigorously? 

Reviewer #1: Yes

Reviewer #2: I Don't Know

3. Have the authors made all data underlying the findings in their manuscript fully available?

Reviewer #1: Yes

Reviewer #2: Yes

4. Is the manuscript presented in an intelligible fashion and written in standard English?

Reviewer #1: Yes

Reviewer #2: Yes

5. Review Comments to the Author

Reviewer #1: The study investigates the effect of three different factors, a. residue placement (surface vs. incorporated), b. residue size (chopped vs. ground sieved) and c. soil type (sand and sandy loam) on barley residue decomposition rates by using a specific modeling approach.

In general, introduction section covers sufficient the background knowledge regarding the objectives of the study. Moreover materials and methods are clear and easily captured by the reader. In this line properties of soils and residue used are properly described by the study. Results and discussion are also adequate. However, more detailed discussion could be strengthen the results and provide the basis for the future work that is needed.

Overall, the manuscript is interesting and provide possible options regarding the effects of residue management in decomposition rates of residues left from barley cultivations. Therefore, I suggest it for publication after minor revision to clarify few parts of the text and increase the potential of the provided information, as follows:

L27. Please add g kg-1 dry residue. Do the same throughout the manuscript.

L70. It would be better to mention that these are conflicting results indicating the effects of other co-factors (e.g. residue properties, native microbial rather than claim that there is no any effect by type of residue placement.

L75-77. “the application of residues in coarser chopped conditions would promote soil aeration to help build healthy soil biota and conserve more water through a consistent breakdown.”. How we can be sure that is the case. Please add references to support it.

L97. to ensure… that. Please modify the sentence, it is too long and the meaning gets confusing.

L101. The study relies on estimates in decomposition rates. Therefore I would suggest to modify from “decomposition” to decomposition rate.

L. 247-254. Figure captions should be at the end of the manuscript followed by figure. Do the same for tables.

Please reform the tables by reducing the lines. See other examples from the studies published in the journal.

Figure 3. Please add st. error or st. dev to the figure to highlight variation of the data.

L322-324. In reasons behind the increased initial decomposition rates I would add the presence of labile organic material (compounds) that fist undergo decomposition by microbial activity. It is mentioned few lines below in the text.

L335-337. This isn’t clear. Please explain further and use references.

L368-369. Please discuss more and provide explanation before you suggest that “Thus, the fine-sized grinding of residue provided no advantage over the chopping of residues.”.

L374-379. Please do the same as above. Strengthen discussion regarding the lack of difference between soil types. Apparently several factors may have been involved. You might discuss the similarity regarding the soil texture due to relatively low clay content or similar low organic matter content (0.9-1.5%)(that may drive aeration-moisture etc.) as well as you could recall similar incubation conditions. On the other hand the existing differences between soils (biological properties are unknown) somehow may mask potential differences in the response of the added residue as you have mentioned. Please elaborate more and use references.

L390-392. Besides the general suggestion for future microbial community analyses you might be more specific by proposing estimates in population of heterotrophic (micro)organisms (fungi and different bacterial phylum) and information regarding their community composition and structure across different soil types (possibly with more profound differentiations in soil properties than the soils you used). For example differences in organic matter content and in texture profile of between fine-textured and sandy soils. Furthermore, field studies are always welcome along with microcosm experimentations to provide information under realistic conditions and crop presence, as you have mentioned in conclusion section. From another perspective it would be interesting to see how the different residue management affect coupling between decomposition and nitrification or denitrification processes. Such information as above may be included in the conclusion section.

L397-398. Indeed. Please reform this sentence by using the term “priming effect”-increased decomposition of the native organic matter, add references.

Reviewer #2: The manuscript "Evaluation of residue management practices on barley residue decomposition" assess how quick barley residue decay depending on size, placement and soil type. These factors are critical in order to understand the nutrient release for subsequent crops and at the same time carbon (C) capture/loss. The authors have carried out an experiment under laboratory conditions, measured C mineralization and thereafter applied the first-order decay kinetics model to calculate time of decomposition. The weakness of this manuscript is that the experiment is done in the laboratory and probably at too high temperatures 25-30°C therefore the results presented in this study may be in discrepancy with similar studies in the field. Moreover, the manuscript lucks discussion about the role of C capture in cereal dominated agricultural systems, particularly how to increase carbon sequestration instead of to lose it. The manuscript needs revision before it could be accepted for publishing.

Introduction

Generally. Please provide information about soil organic carbon (SOC) in Idaho soils. What is average?

Did you have any hypothesis before you started the experiment?

Line 79 Do you mean soil temperature?

Materials and methods

In M&M, there is a need for more detailed information.

Line 105 – What do you mean with ‘’ a common soil series’’? Please, describe in short pre-history of soil you use in the experiment. What was grown before in a sampling location?

Line 109 – You sampled from 0-15 cm depths, why not from 0-20 cm? What is common soil tillage depth?

Table 1 – Please provide in the text how do you calculate soil organic matter (SOM)? You can replace SOM with the loss on ignition (LOI), which you measured.

Line 144 – Please explain why do you choose so high temperature and was it soil or air temperature? How temperature between 25-30 °C can be a constant?

Line 147 How deep was soil lower in a jar? May you provide diameter of the jar and height?

Line 152-159. Please provide more detailed information about:1) where and how did you place a Petri dish – up, down? For how long time? How frequently you changed them?

Line 164-165. Did you add water to jars containing only soil?

Results

‘’Characterization of soils’’ belongs to Materials and methods. Moreover, you have already described this in Table 1 and there is no need to repeat it. Please delay lines 211-220.

Line 222 – How 1 kg barley residue can contain 3875g (3kg 875 g) total carbon (TC)? Did you analyzed C- content or you just calculated? Using your data C:N ratio then is 95:1, if TN is 41g/kg. There is something wrong.

Line 276 – ‘’where’’ written two times.

Line 285 – only ‘’observed’’ or measured/calculated?

Line 295-299. Here and elsewhere too long sentences and sometimes difficult to understand what you mean. Please rephrase it. You should explain in text what you mean with CI. Better use ‘’± ‘’. There is also no need to write ‘’goodness of fit’’, just R2=0.997.

Table 4 Do you need all text under the table? You have described calculations before in M&M.

Discussion

In general manuscript miss discussion about advantages and disadvantages of breakdown barley residue in production and environmental perspective. How real is 25-30 °C in field? How many days with such temperature can you expect?

Line 390 Did you have microbial community analyses? You haven’t mentioned it earlier.

Line 397 At least you mention building up organic matter. You should discuss more about challenges in cereal production.

Refernces

Line 444 and 461. References 10 and 16 are the same.

Supporting information

Lines 516 – 526 can be moved to Materials and methods as it is important information to the reader.

6. PLOS authors have the option to publish the peer review history of their article (what does this mean?). If published, this will include your full peer review and any attached files.

Reviewer #1: No

Reviewer #2: No

---

## [Author Response · Author response to Decision Letter 0]

7 Apr 2020

Dear Editor and Reviewers, 

Thank you for the opportunity to revise our manuscript, “Evaluation of residue management practices on barley residue decomposition.” We would like to thank the reviewers for their positive comments and have made corrections or addressed all of the reviewers’ comments as noted below. We have provided a track-changes version for the editor and Reviewers.

Regards,

Biswanath Dari (Corresponding Authors)

Following changes were made in general in this revised version of the manuscript:

1. The 'Reference' list has been updated with new references as suggested and appropriate. Accordingly, the order of citation in the main text has changed. We did not maintain the track changes in the ‘Reference’ section because there was quite a change and it would have been crowded if we had to do it in track changes. 

2. The sub-section ‘Calculation of CO2-C decomposed’ section has been moved from the ‘Supplementary section to the main text under the ‘Calculation’ section.

3. The sub-section ‘Characteristics of Soils’ section under the ‘Results’ has been deleted as per the suggestion by Reviewer. 

4. Table 4 has been modified as per the suggestion by the reviewer as well as for better representability (we did not keep the track changes here). 

5. The error bars have been added in Figure 4.

Reviewer: 1

General Comments

The study investigates the effect of three different factors, a. residue placement (surface vs. incorporated), b. residue size (chopped vs. ground sieved) and c. soil type (sand and sandy loam) on barley residue decomposition rates by using a specific modeling approach.

In general, introduction section covers sufficient background knowledge regarding the objectives of the study. Moreover materials and methods are clear and easily captured by the reader. In this line, properties of soils and residue used are properly described by the study. Results and discussion are also adequate. However, more detailed discussion could strengthen the results and provide the basis for the future work that is needed.

Overall, the manuscript is interesting and provide possible options regarding the effects of residue management in decomposition rates of residues left from barley cultivations. Therefore, I suggest it for publication after minor revision to clarify few parts of the text and increase the potential of the provided information, as follows:

Response: We would like to thank the reviewers 1 for the positive comments as well as detailed specific comments that we believe have improved the manuscript. The authors have listed the major requested points from Reviewer 1 below and our responses (Red). All minor comments were also addressed as noted by the track changes document.

Specific Comments

L27. Please add g kg-1 dry residue. Do the same throughout the manuscript.

Response: Thank you for pointing this out. Yes, we had calculated everything based on dry residue. Now the word ‘dry residue’ has been added after ‘g kg-1’ throughout the manuscript. 

L70. It would be better to mention that these are conflicting results indicating the effects of other co-factors (e.g. residue properties, native microbial) rather than claim that there is no any effect by type of residue placement.

Response: Thanks for the suggestion. The sentences has been modifies as (Page 4; Line 74-77): ‘The method of residue placement (surface vs. incorporated) and consequently, the contact between the soil and the residue have had conflicting results indicating the effects of other confounding factors (i.e. residue properties, native microbial communities) on the residue breakdown rate in previous research.’

L75-77. “The application of residues in coarser chopped conditions would promote soil aeration to help build healthy soil biota and conserve more water through a consistent breakdown.”. How we can be sure that is the case. Please add references to support it.

Response: We have added the reference (Sims and Frederick, 1970) to support our statement. The statement reads as follows now (Page 4; Line 83-86): ‘In contrast, the application of residues in coarser chopped conditions would result in reduced surface contact between the soil and barley straw which has been claim to slow down the rate of residue breakdown by microbial decomposers [17]’.

L97. To ensure… that. Please modify the sentence, it is too long and the meaning gets confusing.

Response: We have re-written the statement using two sentences as follows (Page 6; Line 110-114): ‘Additionally, the effect of post-harvest management practices of barley residues in irrigated production systems effect on C-mineralization needs to be quantified and understood to ensure sustainable barley production. Achieving sustainable barley production through understanding the dynamics of residue carbon breakdown and residue CO2-C loss nutrient availability (for example, C, N etc.) for subsequent crop production is the focus of this study’.

L101. The study relies on estimates in decomposition rates. Therefore, I would suggest to modify from “decomposition” to ‘decomposition rate’.

Response: We have changed the word ‘decomposition’ to ‘decomposition rate’. 

L. 247-254. Figure captions should be at the end of the manuscript followed by figure. Do the same for tables.

Response: Thank you for the suggestion. However, we have followed the most recent pattern of organizing the manuscript as per our understanding of the PLOS One journal editorial guidelines. We believe the guidelines state that figure caption and tables (with table caption) should be placed after their first mention in the main text, and the figures should be submitted as a separate file. If we have misunderstood the PLOS One journal guidelines we are more than happy to correct this issue.

Figure 3. Please add st. error or st. dev to the figure to highlight variation of the data.

Response: We already have standard errors (SE) for Figures 1 and 2. The value of SE is smaller so it is hard to see in the figures. Now, we have added SE for Figure 3 as well. . 

L322-324. In reasons behind the increased initial decomposition rates, I would add the presence of labile organic material (compounds) that first undergo decomposition by microbial activity. It is mentioned few lines below in the text.

Response: We have modified the sentence and the sentence reads as follows (Page 21; Line 373-375): ‘This is because the maximum decomposition rate occurs in this period due to soil nutrient dynamics, moisture availability (and soil aeration status), and labile organic material that first undergo decomposition by triggering microbial activity’.

L335-337. This isn’t clear. Please explain further and use references.

Response: The sentences has been re-written as follows (Page 21-22; Line 389-394): ‘Enhancing the physical contact between crop residues and soil minerals may have further promoted the residue-mineral interactions, resulting in lower cumulative decomposition in the incorporated treatment than the surface treatment [5,11,37]’. 

L368-369. Please discuss more and provide explanation before you suggest that “Thus, the fine-sized grinding of residue provided no advantage over the chopping of residues.”

Response: We have modified this section and now reads as follows (Page 23; Line 430-437): ‘Differences were observed in the initial decomposition rate through day-5 which afterwards disappeared. In addition, the amount of C that were not mineralized in the initial period decomposition (~ 5 days) remained quite stabilized in the soil for longer duration after initial flush of CO2-C release. Thus, the fine-sized grinding of residue provided no advantage over the chopping of residues in terms N release (and subsequent crop uptake) from decomposing residue for sustainable agricultural production [11]’.

L374-379. Please do the same as above. Strengthen discussion regarding the lack of difference between soil types. Apparently several factors may have been involved. You might discuss the similarity regarding the soil texture due to relatively low clay content or similar low organic matter content (0.9-1.5%) (That may drive aeration-moisture etc.) as well as you could recall similar incubation conditions. On the other hand the existing differences between soils (biological properties are unknown) somehow may mask potential differences in the response of the added residue as you have mentioned. Please elaborate more and use references.

Response: Thanks for the comment. It helps to improve this section of the discussion. Now, we have modified the paragraphs as follows (Page 24; Line 442-454): ‘Our study did not show any significant differences in the cumulative CO2-C release by the two soils (Table 3). These two soils possess similar soil texture, however, these soils were selected as they are representative of the relatively similar soils found in the study region (south-eastern Idaho) (Table 1). Due to their similar soil texture, the CO2-C mineralized from both the soils irrespective of the type of residue added were similar. This might be due to the fact that both soils have lower clay or organic matter content (0.9-1.5%) which has influenced the aeration and moisture status of the soil-residue mixtures [19]. Further, the presence of clay mineral (organo-clay mineral association) controls the magnitude of residue-mineral interactions which was not significantly different between these two soil types as both are relatively low in clay content. Additionally, other associated soil biological factors such as SOM did not vary widely which could have possibly made a difference in microbial populations and nutrient dynamics in these two soil types, and thus, differences in CO2-C release from the soils’.

L390-392. Besides the general suggestion for future microbial community analyses you might be more specific by proposing estimates in population of heterotrophic (micro)organisms (fungi and different bacterial phylum) and information regarding their community composition and structure across different soil types (possibly with more profound differentiations in soil properties than the soils you used). For example, differences in organic matter content and in texture profile of between fine-textured and sandy soils. Furthermore, field studies are always welcome along with microcosm experimentations to provide information under realistic conditions and crop presence, as you have mentioned in conclusion section. From another perspective it would be interesting to see how the different residue management affect coupling between decomposition and nitrification or denitrification processes. Such information as above may be included in the conclusion section.

Response: There are great suggestions and really helpful. As per the suggestions, we have modified the sentence as follows (Page 25; Line 465-470): ‘Further microbial community analyses which estimates the population of heterotrophic microorganisms (e.g. fungi, different bacterial phylum, etc.) and information regarding their community composition and structure (e.g. bacterial:fungal ratio determination using Phospholipid-derived fatty acid; PLFAs [40]) across different soil types might explain the variability in k values for various treatment combinations’. 

We have now modified/added the following statement in the conclusion section as follows (Page 26; Line 494-502): ‘However, barley residue decomposition study under field conditions along with microcosm experimentation in laboratory, and measured at varying times during the crop growing season is warranted to achieve a definite conclusion. Besides, the inclusion of residues in soils with varying SOM content and/or texture profile (e.g. fine- vs. coarse-textured) would be helpful to achieve a definite conclusion. As recommendations are developed, they should consider the advantages and disadvantages of faster or slower residue breakdown under field condition, which is location- and soil- specific. Thus, it would be interesting to see if various residue management practices have coupling effects on soil C (soil C formation and loss) and N cycle (nitrification and denitrification) processes’.

L397-398. Indeed. Please reform this sentence by using the term “priming effect”-increased decomposition of the native organic matter, add references.

Response: Thanks for this great comments. We have modified the sentence and added a sentence. The statements reads as follows (Page 24-25; Line 475-480): ‘However, faster residue mineralization might have a priming effects by increasing the decomposition of native organic matter, thus, reducinge the soil's organic matter buildup [41]. Results indicated that we can achieve a faster C-mineralization rate of for surface-chopped residues mixed with soils under controlled laboratory conditions over a period of 50-d. However, a future study would be warrant to capture the priming effects and it would require labelling of either soil or the residue’.

Reviewer: 2

General Comments

The manuscript "Evaluation of residue management practices on barley residue decomposition" assess how quick barley residue decay depending on size, placement and soil type. These factors are critical in order to understand the nutrient release for subsequent crops and at the same time carbon (C) capture/loss. The authors have carried out an experiment under laboratory conditions, measured C mineralization and thereafter applied the first-order decay kinetics model to calculate time of decomposition. The weakness of this manuscript is that the experiment is done in the laboratory and probably at too high temperatures 25-30°C, therefore, the results presented in this study may be in discrepancy with similar studies in the field. Moreover, the manuscript lacks discussion about the role of C capture in cereal dominated agricultural systems, particularly how to increase carbon sequestration instead of to lose it. The manuscript needs revision before it could be accepted for publishing.

Response: We would like to thank the reviewers 2 for the positive comments, criticism, as well as detailed specific comments that we believe, have improved the manuscript. The authors have listed the major requested points from Reviewer 2 below and our responses (Red). All minor comments were also addressed as noted by the track changes document.

We have re-written the following sentence to explain this further in the sub-section ‘Laboratory incubation experimental approach’ under the section ‘M&M’ as follows (Page 8-9; Line 165-168): ‘A 50-d laboratory incubation experiment was conducted at a constant temperature (25°C) and moisture (60% water-filled pore space). Although we did not mimic the field condition, the experimental set-up we used allowed us to compare our findings with other microcosm studies focused on residue decomposition [5,18]’.

Temperature was a constant factor in the experiment and thus, was not manipulated for the experiment. As a laboratory experiment the temperature was set at an optimal range of both moisture and temperature to determine differences under these conditions similar to previous work (Gilmour et al., 1998, Al Kaisi et al., 2016, 2017, Rogers et al. 2017, etc.). To elaborate, the study was conducted at an air temperature of 25°C under laboratory conditions which is within the range of summer field conditions (June to August) in the study region and in many temperate regions. We aimed to conduct this study with air temperature of 25°C which optimizes the microbial activity and captures the maximum decomposition i.e. C-mineralization (Gilmour et al. 1998). Thus, this temperature condition can be related to how different residue type impact decay constant i.e. k values. 

We have corrected the temperature as a 25°C temperature was maintained in the incubator throughout the experiment. We have corrected this issue in the main text. 

We have tried our best to explain by reasoning the difference in C-mineralization (i.e. residue decomposition) under various soil-residue mixtures in a barley (as a cereal crop) production system. The main purpose of this study was to show the CO2-C release and how it differs as per various barley residue types. We did not intend to measure soil carbon sequestration which was beyond the scope of this study. Our study warrant future studies which could involves evaluating soil carbon capture/carbon storage due different management practices (e.g. tillage). Carbon being captured in the soil is affected by management practices and may influence carbon sequestration. However, post-harvest residues from these cereal crops can limit available N in the soil due to high C: N ratios. To decompose residues with high C: N ratios, growers may attempt different types of tillage. Tillage can affect the amount of residue cover left on the field and is variable depending on management preferences. Each type of tillage management will affect the status of residue cover and the amount of residue-soil contact. For example, when using a moldboard plow for corn/small grain residue, there is only 0 to10 percent of residue cover left in the field (USDA, 2018). In contrast, using a disk (offset, primary >9” spacing) or a chisel will result in 40 to80 percent of residue cover leftover on the soil surface (USDA, 2018). Management of residue cover is ultimately determined by what is available mechanically by the grower.”

Reference: https://www.nrcs.usda.gov/wps/portal/nrcs/detail/national/technical/nra/rca/?cid=nrcs144p2_027241

Introduction

Generally. Please provide information about soil organic carbon (SOC) in Idaho soils. What is average?

Response: The average soil organic carbon content of Idaho soils from previous studies in the predominant agriculture production areas is 2 (±0.1) g kg-1 (Rogers et al. 2019). The detailed information can be found in our other publication at Rogers et al., 2019. We have added the following sentence in the ‘Site Description’ sub-section under the section ‘M&M’ as follows (Page 6; Line 125-127): ‘The average soil organic carbon content measured by loss on ignition (LOI) in the study region was 2 g kg-1 [23]’. 

Comment: Did you have any hypothesis before you started the experiment?

Response: Thanks for your suggestion. Yes we had hypotheses before we started, however, we have chosen to simply state the research questions and objectives instead of hypothesis. We would be more than happy to add the hypotheses, if further needed. 

Line 79. Do you mean soil temperature? 

Response: We have modified the sentence to capture the status of soil temperature as suggested in the reference. We have re-written the sentence as follows (Page 5; Line 87-90): ‘Thus, smaller sized residues may favor C-mineralization and residue breakdown in barley by either increased microbial activities particularly by increasing soil temperature [15] or by filling the soil pore-space in higher magnitude which facilitate decomposition [5]’.

Materials and methods

In M&M: There is a need for more detailed information.

Line 105 – What do you mean with ‘’ a common soil series’’? Please, describe in short pre-history of soil you use in the experiment. What was grown before in a sampling location?

Response: Thanks for the comment. However, we are not clear on this comment as we have already mentioned about the soil series with references such as (Page 6; Line 121-123): ‘Two soil samples (with four field replicates) were collected from a common soil series (Declo Loam; Coarse-loamy, mixed, superactive, mesic Xeric Haplocalcids) in southeastern Idaho [21,22] (Table 1)’. We are not certain on what other info can be added here. Additionally, we have added the reference to the USDA-NRCS soil classification system. I hope it helps. We would be happy to provide more info if applicable.

We have added one sentence on the type of crops grown in the soils before sampling as (Page 6; Line 124-125): ‘Before soil sample collection, the lands were cultivated with barley (Hordeum vulgare, L.)’. 

Line 109 – You sampled from 0-15 cm depths, why not from 0-20 cm? What is common soil tillage depth?

Response: Thanks for this comment. We have added the following sentence along with a reference to explain as follows (Page 6; Line 129-130): ‘Tillage operations can be highly varied in Idaho and other western States. The depth of 0-15 cm is an approximation for shallow tillage operations that are conducted using disc plows [5]’.

Table 1: Please provide in the text how do you calculate soil organic matter (SOM)? You can replace SOM with the loss on ignition (LOI), which you measured.

Response: We have kept the SOM values as measured by LOI methods throughout the manuscript to maintain consistency and avoid any confusion. 

Line 144. Please explain why do you choose so high temperature and was it soil or air temperature? How temperature between 25°C can be a constant?

Response: Please see our responses to the general comment by Reviewer 2 as well. We acknowledge the fact that the study was conducted at a constant air temp of 25°C under laboratory conditions (at an incubator). The goal of the study was to optimizes the microbial activity and captures the maximum decomposition i.e. C-mineralization in the study region (Gilmour et al. 1998). Thus, this temperature condition can be related to how different residue type impact decay constant i.e. k values. We have re-written the following sentence to explain this further in the sub-section ‘Laboratory incubation experimental approach’ under the section ‘M&M’ as follows (Page 8-9; Line 165-168): ‘A 50-d laboratory incubation experiment was conducted at a constant temperature (25°C) and moisture (60% water-filled pore space). Although we did not mimic the field condition, the experimental set-up we used allowed us to compare our findings with other microcosm studies focused on residue decomposition [5,18]’.

Line 147. How deep was soil layer in a jar? May you provide diameter of the jar and height? 

Response: The information are provided as follows (Page 9; Line 174-179): ‘Barley residue (4.1 g) and soil (100 g) were placed in a Mason jar. A 500 mL capacity wide-mouth mason jars with an 86 mm dome lid modified to support a 60-mm petri dish (32) were used in this study. The inner diameter width 76 cm, outer diameter width of 82 cm, and height of 113cm. (The inner diameter width 76 cm, outer diameter width of 82 cm, and height of 113cm)’.

The depth of soil-residue mixture was approximately 35-45 cm from the bottom of the jar. 

Line 152-159. Please provide more detailed information about: 1) where and how did you place a Petri dish – up, down? For how long time? How frequently you changed them? 

Response: Thanks for your comments. We have re-written and added one sentence in main text as follows (Page 9; Line 182): ‘The study was conducted based on incubation methods described by [5]. A statement describing the frequency of change in Petri-dishes are also mentioned in the text already as follows (Page 9; Line 189-190): ‘A new petri dish with 1 M NaOH was used at each time interval after taking the previous petri-dish from individual mason jars’. We would be more than happy to provide further details if needed. The mason jars were constructed such a way so that it allows the Petri dishes to be suspended above the soil.

Line 164-165. Did you add water to jars containing only soil? 

Response: Yes, we have added water to the jars containing soil-residue mixture or soils only to retain soil moisture at optimal levels. This includes both soil with residue added and soil without residue added which were both needed to make accurate calculations of specific losses from residue as opposed to those from the actual soil.

Results

“Characterization of soils”: Belongs to Materials and methods. Moreover, you have already described this in Table 1 and there is no need to repeat it. Please delete lines 211-220.

Response: We have deleted the entire sub-section ‘Characteristics of Soils’ under the ‘Results’ section.

Line 222. How 1 kg barley residue can contain 3875g (3kg 875 g) total carbon (TC)? Did you analyzed C- content or you just calculated? Using your data C:N ratio then is 95:1, if TN is 41g/kg. There is something wrong. 

Response: We apologize for our typo. Now, we have corrected the values in Table 2 (Page 14) and the main texts. 

Line 276. ‘Where’ written two times.

Response: One extra ‘where’ has been deleted. 

Line 285. Only ‘observed’ or measured/calculated?

Response: It would be measured. It has been corrected now. 

Line 295-299. Here and elsewhere too long sentences and sometimes difficult to understand what you mean. Please rephrase it. You should explain in text what you mean with CI. Better use ‘’± ‘’. There is also no need to write ‘’goodness of fit’’, just R2=0.997.

Response: Thanks for the positive note here. We have modified CI into ‘±’ throughout the manuscript including modified Table 4 (Page 19-20) and in the section ‘First-order decay kinetics and half-life of barley residues’ under the section ‘Results’ (Page 17; Line 313-333). We have used only ‘R2’ instead of ‘goodness of fit’ throughout the text and under the section ‘First-order decay kinetics and half-life of barley residues’ (Page 17). We have either re-phrased or split the sentences under the section ‘First-order decay kinetics and half-life of barley residues' (Page 17) for better readability. Please see the modified Table 4. 

Table 4. Do you need all text under the table? You have described calculations before in M&M.

Response: Thanks for the note. We have deleted most of the footnote from under Table 3 (Page 16-17) and Table 4 (Page 19-20) except that are needed which were already explained in the section ‘Calculation’ under the section ‘M&M’. 

Discussion

Comments: In general manuscript miss discussion about advantages and disadvantages of breakdown barley residue in production and environmental perspective. How real is 25-30°C in field? How many days with such temperature can you expect?

Response: Please see our responses for the general comment and comment given on Line 144. We hope it will clarify. We would be happy to provide further details if needed. 

Line 390. Did you have microbial community analyses? You haven’t mentioned it earlier.

Response: We did not perform any microbial community analyses for this study. However, as we indicated our responses to the comment for Line 390-392 by Reviewer 1 on this same comment we believe it could be of benefit in future studies. We hope this will help. 

Line 397. At least you mention building up organic matter. You should discuss more about challenges in cereal production.

Response: The focus of this study was on the decomposition of the actual residue and focus on specific organic matter buildup was beyond the results measured in the study. Additionally, the timeframe of the experiment, and the long duration to influence SOM result in data beyond the scope of the study. However, we agree that soil organic matter would be influenced by the results of the decomposition of the cereal straws but are hesitant to go further than we have with the data that was produced in this laboratory study. We are unclear on what specific challenges the reviewer is interested in and thus, are uncertain of what they are requesting we add without more details.

References

Line 444 and 461. References 10 and 16 are the same.

Response: Thanks for catching the error. The duplicate reference has been deleted from the ‘Reference’ list and the citation order in the main text and ‘Reference’ list has been updated accordingly. 

Supporting information

Lines 516 – 526. Can be moved to Materials and methods as it is important information to the reader.

Response: This section has been moved to the main text in the sub-section ‘Calculation’ under the section ‘M&M’ after the subsection ‘First-order decay constant calculation’.

---

## [Decision Letter · Decision Letter 1]

24 Apr 2020

Evaluation of residue management practices on barley residue decomposition

PONE-D-19-33598R1

Dear Dr. Dari,

We are pleased to inform you that your manuscript has been judged scientifically suitable for publication and will be formally accepted for publication once it complies with all outstanding technical requirements.

With kind regards,

Vassilis G. Aschonitis

Academic Editor

PLOS ONE

Additional Editor Comments (optional):

Reviewers' comments:

Reviewer's Responses to Questions

**Comments to the Author**

1. If the authors have adequately addressed your comments raised in a previous round of review and you feel that this manuscript is now acceptable for publication, you may indicate that here to bypass the “Comments to the Author” section, enter your conflict of interest statement in the “Confidential to Editor” section, and submit your "Accept" recommendation.

Reviewer #1: All comments have been addressed

Reviewer #2: All comments have been addressed

2. Is the manuscript technically sound, and do the data support the conclusions?

Reviewer #1: Yes

Reviewer #2: Yes

3. Has the statistical analysis been performed appropriately and rigorously? 

Reviewer #1: Yes

Reviewer #2: (No Response)

4. Have the authors made all data underlying the findings in their manuscript fully available?

Reviewer #1: Yes

Reviewer #2: Yes

5. Is the manuscript presented in an intelligible fashion and written in standard English?

Reviewer #1: Yes

Reviewer #2: Yes

6. Review Comments to the Author

Reviewer #1: Thank you for your response to my comments.

L75. Please remove “in previous research”.

L83. “soil temperature” isn’t possible discrimination factor between differentially sized residues. In my opinion it would be avoided.

Styles of the Tables used could be changed.

Reviewer #2: The Authors have taken into consideration the comments and have improved the manuscript. This manuscript is now acceptable for publication after minor revision. Pge 9, lines 177-179 the same text is written twice.

7. PLOS authors have the option to publish the peer review history of their article (what does this mean?). If published, this will include your full peer review and any attached files.

Reviewer #1: No

Reviewer #2: No

---

## [Editor Report · Acceptance letter]

28 Apr 2020

PONE-D-19-33598R1 

Evaluation of residue management practices on barley residue decomposition 

Dear Dr. Dari:

I am pleased to inform you that your manuscript has been deemed suitable for publication in PLOS ONE. Congratulations! Your manuscript is now with our production department. 

With kind regards,

on behalf of

Dr. Vassilis G. Aschonitis 

Academic Editor

PLOS ONE